# Health behaviours and mental and physical health status in older adults with a history of homelessness: a cross-sectional population-based study in England

Lee Smith,[1] Nicola Veronese,[2] Guillermo Felipe López-Sánchez,[3] Eloise Moller,[4] James Johnstone,[1] Joseph Firth,[5] Igor Grabovac,[6] Lin Yang,[7] Pinar Soysal,[8] Sarah E Jackson[9]

For numbered affiliations see end of article.

**Correspondence to**
Dr Lee Smith;
lee.smith@anglia.ac.uk

## ABSTRACT

**Objectives** This study compared (1) levels of engagement in lifestyle risk behaviours and (2) mental and physical health status in individuals who have previously been homeless to those of individuals who have not.

**Design** Cross-sectional.

**Participants** Data were from participants (n=6931) of the English Longitudinal Study of Ageing.

**Measures** Participants reported whether they had ever been homeless. We used regression models to analyse associations between homelessness and (1) cigarette smoking, daily alcohol consumption and physical inactivity, adjusting for sociodemographic covariates (age, sex, ethnicity, highest level of education, marital status and household non-pension wealth) and (2) self-rated health, limiting long-standing illness, depressive symptoms, life satisfaction, quality of life and loneliness, adjusting for sociodemographics and health behaviours.

**Results** 104 participants (1.5%) reported having been homeless. Individuals who had been homeless were significantly more likely to be physically inactive (OR 1.62, 95% CI 1.44 to 2.52), report fair/bad/very bad self-rated health (OR 1.75, 95% CI 1.07 to 2.86), have a limiting long-standing illness (OR 2.66, 95% CI 1.65 to 4.30) and be depressed (OR 3.06, 95% CI 1.85 to 5.05) and scored lower on measures of life satisfaction (17.34 vs 19.96, p<0.001) and quality of life (39.02 vs 41.21, p=0.013). Rates of smoking (20.2% vs 15.4%, p=0.436), daily drinking (27.6% vs 22.8%, p=0.385) and loneliness (27.1% vs 21.0%, p=0.080) were also elevated.

**Conclusions** Those who were once homeless have poorer mental and physical health outcomes and are more likely to be physically inactive. Interventions to improve their health and quality of life are required.

## INTRODUCTION

Homelessness is a substantial and growing problem in the UK. The annual homelessness monitor from Crisis and the Joseph Rowntree Foundation showed that in 2015/2016 there were 271000 local authority homelessness

### Strengths and limitations of this study

► To our knowledge, this is the first study to examine health outcomes in those who have transitioned out of homelessness.
► The very small number of participants with a history of homelessness in our sample meant that analyses were underpowered to detect modest differences between groups.
► However, the fact that we observed significant differences in the majority of the outcomes we analysed attests to the strength of these associations.
► Information on time since the period(s) of homelessness was not available so we were unable to evaluate the extent to which recency of homelessness is related to our outcomes of interest.
► Information on what way the participants were homeless (rough sleepers, statutory homeless families) was not available.

case actions in the UK, a rise of 32% since 2009/2010.[1] Being homeless or at risk of homelessness has been shown to have a detrimental impact on mental and physical health.[2] A recent systematic review concluded that people who are homeless are at increased risk of respiratory conditions, depression, anxiety and excess winter mortality, compared with the general population.[3] Homelessness is associated with premature death, with the single homeless at the highest risk with an average age of death at 47 years, some 30 years lower than in the general population.[4] Moreover, the standardised mortality ratios reported for the homeless vary between studies and countries but are typically 2–5 times the age-standardised general population.[5]

Increased morbidity and mortality among the homeless may be driven, at least in part,

by higher levels of engagement in lifestyle risk behaviours. Data from the USA indicate that while 19.8% of adult Americans smoke, smoking prevalence is over 70% among those who are homeless.[6–9] Levels of physical activity are also low among the homeless. In a Danish study, approximately 70% of homeless individuals reported no participation in any form of exercise.[10] High levels of alcohol consumption and drug use are also common among this population.[11]

Tackling homelessness is an urgent priority, and targeted policies have been actioned in the UK to rehouse those who are homeless. The Homelessness Directorate was established in 2002 in order to assist local authorities in tackling homelessness.[12] Strategies focus on preventing the need for people to sleep rough in the first place, as well as supporting people to move on from homelessness by helping them to address their needs, improving access to health and substance misuse services, and helping them rebuild their lives through education, training and employment.[13] A number of UK charities (eg, Crisis, Shelter England, The Single Homeless Project) also work to support people who are homeless in acquiring a home and entering back into employment. With such policies and charities in operation, a significant number of individuals are able to transition out of homelessness.

While the evidence base on the health risks associated with homelessness is growing, to our knowledge, no studies have explored what happens to the health and well-being of people when they are no longer homeless. Given that lifestyle behaviours tend to track over the life course,[14] and early life exposures can have a substantial impact on later-life health outcomes,[15] it seems likely that the health risks associated with homelessness may persist, at least to some extent, beyond the period of homelessness. The present study aimed to investigate this through a comparison of (1) levels of engagement in lifestyle risk behaviours and (2) mental and physical health status in individuals who have previously spent a period of time in their lives as homeless with those of individuals who have never been homeless, in a population-based sample of older adults living in England. Specifically, we aimed to address the following research questions:

1. To what extent do individuals with a history of homelessness differ from those who have never been homeless with regard to smoking status, alcohol intake and level of physical activity, adjusting for relevant sociodemographic characteristics?
2. To what extent do individuals with a history of homelessness differ from those who have never been homeless with regard to self-rated health, limiting long-standing illness, depressive symptoms, life satisfaction, quality of life (QoL) and loneliness, adjusting for relevant sociodemographic characteristics and health behaviours?

We hypothesised that individuals who had previously been homeless would have a higher prevalence of lifestyle risk behaviours and an unfavourable mental and physical health profile compared with those who had never been homeless.

## METHODS

### Study population

Data were from the English Longitudinal Study of Ageing (ELSA), a nationally representative longitudinal panel study of men and women aged 50 and older living in households across England.[16] The study began in 2002, with subsequent rounds of data collection at 2-year intervals via computer-assisted personal interview and self-completion questionnaires. Wave 3 (2006/2007) included a life history questionnaire, which gathered detailed information about important events that occurred in the participants' lives, including whether they had ever been homeless. Of the 9771 participants interviewed in wave 3 of ELSA, 7855 (80.4%) completed the life history questionnaire. We excluded 924 participants (11.8%) with missing data on homelessness or sociodemographic covariates, leaving a final sample for analysis of 6931 men and women.

### Patient and public involvement

Patients were not involved in the design of this study.

### Measures

#### History of homelessness

Participants were asked whether they had ever been homeless for 1 month or more (yes/no).

#### Health behaviours

Smoking status was assessed with the question 'Do you smoke cigarettes at all nowadays' (yes/no).

Frequency of alcohol intake over the past 12 months was reported on an eight-point scale from 'not at all in the last 12 months' to 'almost every day'. We dichotomised responses to distinguish between participants drinking almost every day ('daily drinking') versus less than this.

Physical activity was assessed with three items that asked respondents about the frequency with which they took part in vigorous, moderate and low-intensity activities (more than once a week, once a week, 1–3 times a month, hardly ever/never).[17] Responses were dichotomised as follows: inactive (no moderate/vigorous activity on a weekly basis) versus active (moderate or vigorous activity at least once a week).

#### Health and well-being

Self-rated health was assessed using a single item: 'Would you say your health is… very good/good/fair/bad/very bad?' We analysed the proportion of individuals rating their health as fair, bad or very bad, as is commonly done in analyses of this variable.[18 19]

Limiting long-standing illness was assessed with two questions: (1) 'Do you have any long-standing illness, disability, or infirmity? By long-standing I mean anything that has troubled you over a period of time or that is likely to affect you over a period of time.' If they responded yes, they were asked (2) 'Does this illness or disability limit your activities in any way?' Affirmation of a long-standing illness and any form of limitation classified the participant as having a limiting long-standing illness.

Depressive symptoms were assessed with an eight-item version of the Center for Epidemiologic Studies Depression Scale,[20] a scale highly validated for use in older adults.[21] This asks about feelings over the last week (eg, 'Over the last week have you felt sad'), with binary response options (1=yes, 0=no). Positively framed items were reverse scored. Data were dichotomised using an established cut-off, with a score of 4 or higher indicating significant symptomatology.[21]

Life satisfaction was assessed with the Satisfaction With Life Scale,[22] which asks respondents to rate the extent to which they agree with five statements: 'In most ways my life is close to my ideal'; 'The conditions of my life are excellent'; 'I am satisfied with my life'; 'So far I have got the important things I want in life'; 'If I could live my life again, I would change almost nothing' on a scale from 0 (strongly disagree) to 6 (strongly agree). Responses are summed to produce a total score between 0 and 30, with higher scores indicating greater life satisfaction.

QOL was assessed with the Control, Autonomy, Self-relisation, Pleasure (CASP) -19,[23] a scale designed to measure QOL in older people. Items cover four domains of QOL; control (eg, 'I feel that what happens to me is out of my control'), autonomy (eg, 'My health stops me from doing things I want to do'), self-realisation (eg, 'I feel that life is full of opportunities'), and pleasure (eg, 'I enjoy being in the company of others'). Respondents are asked how often each statement applies to them (often=0, sometimes=1, not often=2, never=3). Positively worded items are reverse scored so that a higher total score indicates higher QOL (range: 0–57).

Loneliness was measured using a three-item short form of the Revised University of California Los Angeles Loneliness Scale.[24] Participants were asked: 'How often do you feel you lack companionship?'(hardly ever or never=1, some of the time=2, often=3). Scores ranged from 3 to 9, with higher scores indicating greater loneliness. They were dichotomised at ≥6 vs <6 to indicate high versus low loneliness.[25]

### Sociodemographic covariates
Interviewers collected information on age, sex, ethnicity, the highest level of education, marital status and wealth. For these analyses, ethnicity was categorised as white or non-white. We classified education as low (no formal qualifications), intermediate (up to degree) or high (degree or higher). Marital status was categorised as married or unmarried (never married, divorced or widowed). Wealth was categorised into five equal groups of net total non-pension wealth measured at the benefit unit level (a benefit unit is a couple or single person along with any dependent children they might have) across all ELSA participants who took part in wave 3. Wealth has been identified as a particularly appropriate indicator of SES in this age group.[26]

### Statistical analysis
All analyses were conducted using SPSS V.24. Data were weighted to correct for sampling probabilities and for differential non-response and to calibrate back to the 2011 National Census population distributions for age and sex. The weights accounted for the differential probability of being included in wave 3 of ELSA and for non-response to the life history interview.

Differences in sociodemographic characteristics of the groups who did and did not report a history of homelessness were tested using independent t-tests for continuous variables and Pearson's $X^2$ tests for categorical variables. We used binary logistic regression to analyse associations between history of homelessness and cigarette smoking, daily drinking and physical inactivity, adjusting for sociodemographic covariates. We then used linear regression (for continuous outcomes) and binary logistic regression (for categorical outcomes) to analyse associations between history of homelessness and health and well-being, adjusting for sociodemographics and health behaviours. In all models, the reference category was the group without a history of homelessness. A p<0.05 was used to indicate statistical significance.

## RESULTS
### Sample characteristics
Of the 6931 participants in our sample, 104 (1.5%) reported having been homeless for 1 month or more and 6827 (98.5%) had never been homeless for 1 month or more. Sample characteristics in relation to history of homelessness are summarised in table 1. On average, participants who had been homeless were significantly younger than those who had not been homeless (60.9 vs 65.7 years) and a greater proportion were non-white (6.6% vs 3.1%), unmarried (54.3% vs 33.8%) and from the lowest quintile of wealth (44.3% vs 18.9%). A marginally higher proportion of the group who had been homeless were male (53.8% vs 46.7%) although the difference was not statistically significant (p=0.149). There was no significant difference between groups in the highest level of education achieved.

### History of homelessness and health behaviours
Associations between history of homelessness and health behaviours are shown in table 2. After adjustment for age, sex, ethnicity, education, marital status and wealth, participants who had been homeless had 1.62 times higher odds (95% CI 1.44 to 2.52) of being inactive than those who had not been homeless (30.7% vs 23.0%, p=0.031). Rates of smoking (20.2% vs 15.4%) and daily drinking (27.6% vs 22.8%) were also higher in the group who had been homeless, but differences were not statistically significant.

### History of homelessness and health and well-being
Associations between history of homelessness and health and well-being are summarised in table 3. After adjustment for sociodemographics and health behaviours,

**Table 1** Sample characteristics in relation to history of homelessness

| | Had not been homeless (n=6827)* | Had been homeless (n=104) | P value |
|---|---|---|---|
| Age (years), mean (SD) | 65.74 (10.64) | 60.94 (8.32) | <0.001 |
| Sex | | | |
| Men | 46.7 | 53.8 | 0.149 |
| Women | 53.3 | 46.2 | – |
| Ethnicity | | | |
| White | 96.9 | 93.4 | 0.042 |
| Non-white | 3.1 | 6.6 | – |
| Highest level of education | | | |
| No qualifications | 32.4 | 27.4 | 0.541 |
| Below degree | 52.2 | 56.6 | – |
| Degree or higher | 15.5 | 16.0 | – |
| Marital status | | | |
| Married | 66.2 | 45.7 | <0.001 |
| Unmarried | 33.8 | 54.3 | – |
| Wealth quintile | | | |
| 1 (poorest) | 18.9 | 44.3 | <0.001 |
| 2 | 19.5 | 13.2 | – |
| 3 | 20.6 | 14.2 | – |
| 4 | 20.0 | 16.0 | – |
| 5 (richest) | 21.0 | 12.3 | – |

All figures are weighted for sampling probabilities and differential non-response.
Values are percentages unless otherwise stated.
*Unweighted sample sizes.

**Table 2** Associations between history of homelessness and health behaviours

| | Had not been homeless | Had been homeless | P value |
|---|---|---|---|
| Smoking | | | |
| % (SE) | 15.4 (0.4) | 20.2 (3.3) | – |
| OR (95% CI) | 1.00 (Ref) | 1.21 (0.75 to 1.94) | 0.436 |
| Daily drinking | | | |
| % (SE) | 22.8 (0.5) | 27.6 (4.2) | – |
| OR (95% CI) | 1.00 (Ref) | 1.31 (0.77 to 2.21) | 0.321 |
| Physical inactivity | | | |
| % (SE) | 23.0 (0.5) | 30.7 (3.7) | – |
| OR (95% CI) | 1.00 (Ref) | 1.62 (1.04 to 2.52) | 0.031 |

All figures are weighted for sampling probabilities and differential non-response, and are adjusted for age, sex, ethnicity, education, marital status and wealth.
SE, SE error.

compared with the group who had not been homeless, the group who had been homeless had 1.75 times higher odds (95% CI 1.07 to 2.86) of reporting fair/bad/very bad self-rated health (41.9% vs 30.5%, p=0.025), 2.66 times higher odds (95% CI 1.65 to 4.30) of reporting a limiting long-standing illness (55.8% vs 33.5%, p<0.001) and 3.06 times higher odds (95% CI 1.85 to 5.05) of depressive symptoms (33.3% vs 13.0%, p<0.001). The group who had been homeless also scored lower on average on measures of life satisfaction (17.34 vs 19.96, p<0.001) and QOL (39.02 vs 41.21, p=0.013). The rate of loneliness (27.1% vs 21.0%) was higher in the group who had been homeless but this difference did not reach statistical significance (p=0.110).

## DISCUSSION

In the present analyses, a total of 104 participants reported having been homeless for 1 month or more. Those who reported a history of homelessness had significantly higher odds of physical inactivity than those who had not

been homeless and were more likely to smoke and drink daily but these did not reach significance. Importantly, those who had reported being homeless had a higher odds of reporting fair/bad/very bad self-rated health, limiting long-standing illness and depressive symptoms and scored lower on measures of life satisfaction and QOL. Taken together, these data suggest that people who transition out of homelessness may be at increased risk of partaking in unhealthy behaviour and suffer poorer mental and physical health.

The finding that those who were previously homeless were more likely to be inactive than those who were not is of importance. Indeed, sustained and regular participation in physical activity can aid in the prevention against, and improve the profile of, non-communicable diseases—including those in relation to both physical (eg, cardiorespiratory[27]) and mental (eg, anxiety and depression[28][29]) health, both of which are common in homeless populations.[3] Moreover, similar health profiles were observed in the present manuscript in a population who has transitioned from homelessness. Literature suggests that levels of physical activity track across the life course.[30] Importantly, those who are homeless have critically low levels of physical activity. For example, in a Danish study, approximately 70% of the homeless reported no participation in any form of exercise.[10] This low level of physical activity is potentially tracking through the transition from homelessness.

The novel finding that people who have ever been homeless are at increased risk of adverse physical and mental health outcomes is important. This suggests that the transition from homelessness is not enough to bring the health profile of this population in alignment with the general public. There are several factors that may

**Table 3** Associations between history of homelessness and health and well-being

| | Had not been homeless | Had been homeless | P value |
|---|---|---|---|
| **Fair/bad/very bad self-rated health** | | | |
| % (SE) | 30.5 (0.6) | 41.9 (4.5) | – |
| OR (95% CI) | 1.00 (Ref) | 1.75 (1.07 to 2.86) | 0.025 |
| **Limiting long-standing illness** | | | |
| % (SE) | 33.5 (0.6) | 55.8 (4.7) | – |
| OR (95% CI) | 1.00 (Ref) | 2.66 (1.65 to 4.30) | <0.001 |
| **Depressive symptoms above threshold** | | | |
| % (SE) | 13.0 (0.4) | 33.3 (3.5) | – |
| OR (95% CI) | 1.00 (Ref) | 3.06 (1.85 to 5.05) | <0.001 |
| **Life satisfaction** | | | |
| Mean score (SE) | 19.96 (0.1) | 17.34 (0.7) | – |
| Coeff. (95% CI) | Ref | −2.78 (−4.18 to −1.37] | <0.001 |
| **Quality of life** | | | |
| Mean score (SE) | 41.21 (0.1) | 39.02 (0.9) | – |
| Coeff. (95% CI) | Ref | −2.25 (−4.03 to −0.47] | 0.013 |
| **High loneliness** | | | |
| % (SE) | 21.0 (0.5) | 27.1 (4.2) | – |
| OR (95% CI) | 1.00 (Ref) | 1.50 (0.91 to 2.47) | 0.110 |

All figures are weighted for sampling probabilities and differential non-response and adjusted for age, sex, ethnicity, education, marital status, wealth, smoking status, alcohol intake and physical activity.
Possible scores on the quality of life scale range from 0 to 57 and on life satisfaction scale range from 0 to 30.
SE, SE error; coeff, coefficient.

account for the observed disparity in health. First, depression is prevalent among the homeless community[31] and is a highly recurrent disorder,[32] thus is likely to reoccur after the transition out of homelessness, with significant personal consequences.[32] Low mood may be partly driving the observed negative associations with life satisfaction, QOL and limiting long-standing illness.[33]

Second, people who are homeless are susceptible to multiple health complications. Chronic hepatitis C and coinfections are common among the homeless population.[34] Other conditions that are prevalent among the homeless include tuberculosis, uncontrolled asthma and dermatological infestations.[35] These problems are compounded by high rates of drug and alcohol abuse and together likely contribute to poorer self-rated health, limiting long-standing illness and lower QOL across the lifespan.[36 37]

Interestingly, while differences in health outcomes (self-rated health, limiting long-standing illness, depressive symptoms, life satisfaction and QOL) between the present sample (ex-homeless) and those who have not been homeless were significant, the magnitude of the associations was smaller than has been documented in previous studies.[38] This may be owing to a degree of 'recovery' from homelessness. It may also be an artefact of the type of homelessness. The majority of the present sample who had experienced homelessness may have

been 'statutory homeless' where health outcomes are likely better than rough sleeping.

To our knowledge, this is the first study to examine health outcomes in those who have transitioned out of homelessness. While these findings are important for advancing the evidence base in this area, they should be considered in light of a couple of limitations. The very small number of participants with a history of homelessness in our sample meant that analyses were underpowered to detect modest differences between groups. However, the fact that we observed significant differences in the majority of the outcomes we analysed attests to the strength of these associations. Nevertheless, future research using larger samples is required to confirm or refute our findings. While we adjusted for a wide range of sociodemographic and behavioural covariates, it is possible that the results could be explained by residual confounding by unmeasured variables—that is, the group reporting a history of homelessness were deprived in ways that were not reflected in the existing variables. Information on time since the period(s) of homelessness was not available so we were unable to evaluate the extent to which recency of homelessness is related to our outcomes of interest. It is possible that participants who reported a history of homelessness had transitioned out of homelessness many years or even decades prior. In addition, information on type of homelessness was not available. It

is, therefore, unknown whether those who reported once being homeless were 'statutory homeless', lived on the streets, stayed in a shelter, abandoned building or vehicle. Type of homelessness may have varying influences on health and behaviour. It is plausible to assume that those who are rough sleepers (living on streets, abandoned buildings or vehicles) are at a higher risk of poor health, for example, owing to exposure to cold weather and wet conditions or lack of access to essential facilities such as bathrooms. However, those who are rough sleepers are much more likely to be male (86% male)[1] and a relatively large proportion of our sample who were once homeless were female (46.7%). It may be that the present sample is not representative of the wider homeless population (or at least rough sleepers) in the UK. Future research to tease out the influence of type of previous homelessness on health/behaviour outcomes is required. Finally, ELSA does not collect data on those currently homeless, and thus, it was not possible to have a 'currently homeless' category in the present analyses. Given that we did not observe significant differences in some outcomes previously demonstrated to differ between currently homeless and housed populations (eg, smoking), it might be the case that those who manage to transition out of homelessness are able to offset some of the increased risk associated with having been homeless. Future research may wish to compare those never homeless, those currently homeless and those previously homeless to gain a deeper insight.

## CONCLUSION

In conclusion, the present results indicate that older adults in England who have previously been homeless are more likely to be physically inactive and have poorer mental and physical health outcomes than those who have never been homeless.

A continued initiative to tackle homelessness itself is important. It is also crucial to consider that even those who have transitioned from homelessness continue to be at much higher risk of poor health and well-being. Therefore, continued monitoring and targeted interventions are required to improve health outcomes and QOL in this population. Such interventions may wish to consider lifestyle risk behaviours to improve mental and physical health status.

**Author affiliations**
[1]The Cambridge Centre for Sport and Exercise Sciences, Anglia Ruskin University, Cambridge, UK
[2]National Research Council, Neuroscience Institute, Aging Branch, Padova, Italy
[3]Faculty of Sport Sciences, University of Murcia, Murcia, Spain
[4]The Single Homeless Project, London, UK
[5]NICM Health Research Institute, Western Sydney University, Sydney, Australia
[6]Department of Social and Preventive Medicine, Centre for Public Health, Medical University of Vienna, Vienna, Austria
[7]Cancer Epidemiology and Prevention Research, Alberta Health Services, Calgary, Canada
[8]Department of Geriatric Medicine, Faculty of Medicine, Bezmialem Vakif University, Istanbul, Turkey
[9]Department of Behavioural Science and Health, University College London, London, UK

**Contributors** Study concept and design: LS and SEJ. Analysis and interpretation of data: LS and SEJ. Drafting of the manuscript: LS, NV, GFL-S, EM, JJ, JF, IG, LY, PS and SEJ. Critical revision of the manuscript for important intellectual content: LS, NV, GFL-S, EM, JJ, JF, IG, LY, PS and SEJ.

**Funding** The authors have not declared a specific grant for this research from any funding agency in the public, commercial or not-for-profit sectors.

**Competing interests** None declared.

**Patient consent for publication** Obtained.

**Ethics approval** Ethical approval was obtained from the National Research Ethics Service.

**Provenance and peer review** Not commissioned; externally peer reviewed.

**Data sharing statement** No additional data are available.

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
