## [Reviewer comments · BMJ Open]

ARTICLE DETAILS

TITLE (PROVISIONAL)	Health behaviours and mental and physical health status in older adults with a history of homelessness: a cross-sectional population-based study in England
AUTHORS	Smith, Lee; Veronese, Nicola; López-Sánchez, Guillermo Felipe; Moller, Eloise; Johnstone, James; Firth, Joseph; Grabovac, Igor; Yang, Lin; Soysal, Pinar; Jackson, Sarah

VERSION 1 - REVIEW

REVIEWER	Ramón Chacón-Cuberos University of Almería (Spain)
REVIEW RETURNED	28-Nov-2018

GENERAL COMMENTS	In relation to the article entitled "Health behavior and mental health and physical status in older adults with a history of homelessness: a population-based study", indicate that the study presents a focus of great interest for the line of the journal. This is an important paper and among the first to study the health of those who have transitioned from homelessness. Indeed, this is a population for which research is scarce. However, some minor modifications are required before it can be accepted for publication: - The introduction of the article is adequate, addresses the focus of the problem and uses current references in its majority.- If possible, it would be interesting for the authors to establish the research question at the end of the introduction (before the objectives) as well as their main study hypotheses.- The material and method is perfectly developed. A representative sample is used, the instruments are adequately described and the statistical analysis is correct.- The results are described properly and the tables are self-explanatory. I found the analysis technique very successful, using binary logistic regression and linear regression depending on the type of variables to establish the model.- The authors can expand on the discussion with a couple of sentences on how we are able to intervene in this population and the authors should state some key directions for future research.- The conclusions are well written and concise. If the authors have been able to finally add hypothesis and research question, it is recommended that they be answered in this section.
---

REVIEWER	Dan Lewer University College London, UK
REVIEW RETURNED	15-Jan-2019

GENERAL COMMENTS

This study is valuable because it focuses on a group that has received very limited attention - people who have 'recovered' from homelessness. Like the authors, I'm not aware of any existing studies into the health of this group. There are important limitations to the data - particularly that we don't know when the participants were homeless, and we don't know what they mean by 'homelessness', which can mean a wide range of things.

My comments are:

Strengths & limitations (p3)

- Worth mentioning limitations that (a) we don't know when the participants were homeless; (b) we don't know in what way the participants were homeless (e.g. if they were rough sleepers or statutory homeless families).
- I'm not sure that the data demonstrates the long-lasting impact of homelessness on health, as the design is cross-sectional and we don't know whether the homelessness preceded the poor health outcomes.

Introduction

- (p4) - average age of death is 47. This is not surprising and not necessarily concerning, as there are few homeless people at older ages so the average age of death is inevitably young. The standardised mortality ratio might be a more insightful measure. The recent Lancet review provides some references - [https://doi.org/10.1016/S0140-6736\(17\)31869-X](https://doi.org/10.1016/S0140-6736(17)31869-X).

Methods

- History of homelessness and health & wellbeing. I would suggest providing two estimates: (a) adjusted for age, sex, ethnicity, education [+ potentially marital status and wealth], and (b) these variables plus the health behaviours. This is because the health behaviours may lie on the causal pathway between homelessness and health status. I'm not sure which group marital status and wealth fall into.
- (p7) - I'd suggest shortening the definition of the wealth variable.

Discussion

- (p12) - a key limitation is that we don't know what 'type' of homelessness the participants are reporting. For example, homelessness may mean rough sleeping, or it may mean that the participants are homeless in a statutory way (i.e. the local authority has a responsibility to find housing for them, but they may still be living in a private household, albeit insecure). The ever-homeless group is 54% female and this is different to the profile of rough sleepers, who are 80% male. This seems to suggest that some of the sample are not rough sleepers - though the profile of homeless people may have changed over time.
- I think it may be worth mentioning that residual confounding could explain the results. The 104 people reporting past-homelessness may just be deprived in ways that are not reflected in the existing variables.
- I would suggest comparing the results to studies of currently homeless people. While the ever-homeless group does have worse health than the never-homeless group, the results are much less stark. I think this is a combination of (a) 'recovery' from homelessness; (b) this group typically having a less severe type of homelessness than participants in studies of currently-homeless

	people; (c) residual confounding - the study is probably selecting a group that is deprived in ways not captured by the study variables.
--	--

VERSION 1 – AUTHOR RESPONSE

Reviewer: 1

Reviewer Name: Ramón Chacón-Cuberos

Institution and Country: University of Almería (Spain) Please state any competing interests or state 'None declared': None declared

Please leave your comments for the authors below In relation to the article entitled "Health behavior and mental health and physical status in older adults with a history of homelessness: a population-based study", indicate that the study presents a focus of great interest for the line of the journal. This is an important paper and among the first to study the health of those who have transitioned from homelessness. Indeed, this is a population for which research is scarce.

However, some minor modifications are required before it can be accepted for publication:

- The introduction of the article is adequate, addresses the focus of the problem and uses current references in its majority.
- If possible, it would be interesting for the authors to establish the research question at the end of the introduction (before the objectives) as well as their main study hypotheses.

Response: Thank you we have now incorporated the research question into the manuscript where requested.

“Specifically, we aimed to address the following research questions:

1. To what extent do individuals with a history of homelessness differ from those who have never been homeless with regard to smoking status, alcohol intake, and level of physical activity, adjusting for relevant sociodemographic characteristics?
2. To what extent do individuals with a history of homelessness differ from those who have never been homeless with regard to self-rated health, limiting long-standing illness, depressive symptoms, life satisfaction, quality of life, and loneliness, adjusting for relevant sociodemographic characteristics and health behaviours?

We hypothesised that individuals who had previously been homeless would have a higher prevalence of lifestyle risk behaviours and an unfavourable mental and physical health profile compared with those who had never been homeless.”

- The material and method is perfectly developed. A representative sample is used, the instruments are adequately described and the statistical analysis is correct.
- The results are described properly and the tables are self-explanatory. I found the analysis technique very successful, using binary logistic regression and linear regression depending on the type of variables to establish the model.
- The authors can expand on the discussion with a couple of sentences on how we are able to intervene in this population and the authors should state some key directions for future research.

Response: Thank you for this important comment we have incorporated suggestions for future research throughout the discussions and the following text in the section “conclusion.”

“Such interventions may wish to consider lifestyle risk behaviours to improve mental and physical health status.”

- The conclusions are well written and concise. If the authors have been able to finally add hypothesis and research question, it is recommended that they be answered in this section.

Response: Thank you we have now done this.

“In conclusion, the present results indicate that older adults in England who have previously been homeless are more likely to engage in lifestyle risk behaviours and have poorer mental and physical health outcomes than those who have never been homeless.”

Reviewer: 2

Reviewer Name: Dan Lewer

Institution and Country: University College London, UK Please state any competing interests or state 'None declared': None declared

Please leave your comments for the authors below This study is valuable because it focuses on a group that has received very limited attention - people who have 'recovered' from homelessness. Like the authors, I'm not aware of any existing studies into the health of this group. There are important limitations to the data - particularly that we don't know when the participants were homeless, and we don't know what they mean by 'homelessness', which can mean a wide range of things.

Response: Thank you for your positive review.

My comments are:

Strengths & limitations (p3)

- Worth mentioning limitations that (a) we don't know when the participants were homeless; (b) we don't know in what way the participants were homeless (e.g. if they were rough sleepers or statutory homeless families).

- I'm not sure that the data demonstrates the long-lasting impact of homelessness on health, as the design is cross-sectional and we don't know whether the homelessness preceded the poor health outcomes.

Response: Thank you we have already stated the following: “• Information on time since the period(s) of homelessness was not available so we were unable to evaluate the extent to which recency of homelessness is related to our outcomes of interest.” We have deleted the point in relation to long-lasting impact and included the following:

“• Information on what way the participants were homeless (rough sleepers, statutory homeless families) was not available.”

Introduction

- (p4) - average age of death is 47. This is not surprising and not necessarily concerning, as there are few homeless people at older ages so the average age of death is inevitably young. The standardised mortality ratio might be a more insightful measure. The recent Lancet review provides some references - [https://doi.org/10.1016/S0140-6736\(17\)31869-X](https://doi.org/10.1016/S0140-6736(17)31869-X).

Response: thank you your comment. The number of older adults who are homeless is on the rise and in the UK and it was estimated that up to 42,000 older people are unofficially homeless in England and Wales https://www.ageuk.org.uk/globalassets/age-uk/documents/reports-and-publications/late_life_uk_factsheet.pdf. Moreover, the Single Homeless Project the largest homeless charity in central London has over 1000 patrons aged over 55 years.

Methods

- History of homelessness and health & wellbeing. I would suggest providing two estimates: (a) adjusted for age, sex, ethnicity, education [+ potentially marital status and wealth], and (b) these variables plus the health behaviours. This is because the health behaviours may lie on the causal pathway between homelessness and health status. I'm not sure which group marital status and wealth fall into.

Response: We appreciate this suggestion, and have given it some thought. However, given that the only health behaviour that differs significantly according to homeless history now we have corrected our analyses with the inclusion of age is physical inactivity, and this could well be a consequence of poor health, we have chosen not to overcomplicate the analyses and the interpretation of causal pathways and have retained just one fully adjusted model for each health and wellbeing outcome of interest.

- (p7) - I'd suggest shortening the definition of the wealth variable.

Response: We have done this.

Discussion

- (p12) - a key limitation is that we don't know what 'type' of homelessness the participants are reporting. For example, homelessness may mean rough sleeping, or it may mean that the participants are homeless in a statutory way (i.e. the local authority has a responsibility to find housing for them, but they may still be living in a private household, albeit insecure). The ever-homeless group is 54% female and this is different to the profile of rough sleepers, who are 80% male. This seems to suggest that some of the sample are not rough sleepers - though the profile of homeless people may have changed over time.

Response: Thank you we have included this into our section "limitations".

"Information on type of homelessness was not available. It is therefore unknown whether those who reported once being homeless were "rough sleepers" or "statutory homeless". Type of homelessness may have varying influences on health/ behaviour and future research to tease out the influence of type of previous homelessness on these outcomes is required."

- I think it may be worth mentioning that residual confounding could explain the results. The 104 people reporting past-homelessness may just be deprived in ways that are not reflected in the existing variables.

Response: We have added a note on this to the limitations:

"While we adjusted for a wide range of sociodemographic and behavioural covariates, it is possible that the results could be explained by residual confounding by unmeasured variables – i.e. the group reporting a history of homelessness were deprived in ways that were not reflected in the existing variables."

- I would suggest comparing the results to studies of currently homeless people. While the ever-homeless group does have worse health than the never-homeless group, the results are much less stark. I think this is a combination of (a) 'recovery' from homelessness; (b) this group typically having a

less severe type homelessness than participants in studies of currently-homeless people; (c) residual confounding - the study is probably selecting a group that is deprived in ways not captured by the study variables.

Response: We agree with the limitations that you point out which we have mentioned in our manuscript, we have made greater comparison to studies of the homeless throughout the manuscript and also incorporated the following into the limitations:

“ELSA does not collect data on those currently homeless and thus it was not possible to have a “currently homeless” category in the present analyses. Given that we did not observe significant differences in some outcomes previously demonstrated to differ between currently homeless and housed populations (e.g. smoking), it might be the case that those who manage to transition out of homelessness are able to offset some of the increased risk associated with having been homeless. Future research may wish to compare those never homeless, those currently homeless, and those previously homeless to gain a deeper insight.”

VERSION 2 – REVIEW

REVIEWER	Ramón Chacón Cuberos University of Granada (Spain)
REVIEW RETURNED	14-Mar-2019

GENERAL COMMENTS	After reviewing the manuscript entitled "Health behaviours and mental and physical health status in older adults with a history of homelessness: a populationbased study", indicate that the authors have made the most suggested changes. Likewise, it is noticeable that the manuscript is well structured, written with a high quality and the pertinent statistical analyzes are used. In addition, it presents a topic of interest which is adjusted to the editorial line of the journal, so its publication is recommended.
--

REVIEWER	Dan Lewer University College London
REVIEW RETURNED	13-Mar-2019

GENERAL COMMENTS	Many thanks for the opportunity to re-review this manuscript. The changes have improved it. Some of my original comments could be more fully addressed:  1. The fact that there are more homeless people in older age groups does not make the average age of death a useful statistic. 2. The manuscript could benefit from a clearer discussion of the limited insight into the type of homelessness. I gave 'rough sleeping' and 'statutory homelessness' as two examples - but there is a wide spectrum and no clear-cut categories. I don't think the ELSA ex-homeless participants are representative of the group we typically think of as 'homeless' (probably people who are rough sleeping or living in hostels) since the majority are female. 3. There is some discussion about the health risks of homelessness, but I think the point could be made more clearly that the difference between ex-homeless and others in this study is much smaller than between current homeless and housed
--

	people in other studies - which have found extremely stark differences in health. This may be due to a degree of 'recovery' from homelessness, but also that the type of homelessness experienced by people in this study is different and less 'severe'.
--	---

VERSION 2 – AUTHOR RESPONSE

Reviewer(s)' Comments to Author:

Reviewer: 2

Reviewer Name: Dan Lewer

Institution and Country: University College London Please state any competing interests or state 'None declared': None

Please leave your comments for the authors below Many thanks for the opportunity to re-review this manuscript. The changes have improved it. Some of my original comments could be more fully addressed:

1. The fact that there are more homeless people in older age groups does not make the average age of death a useful statistic.

Response: we have now included the following.

“Moreover, the standardised mortality ratios reported for the homeless vary between studies and countries but are typically 2–5 times the age-standardised general population.[5]”

2. The manuscript could benefit from a clearer discussion of the limited insight into the type of homelessness. I gave 'rough sleeping' and 'statutory homelessness' as two examples - but there is a wide spectrum and no clear-cut categories. I don't think the ELSA ex-homeless participants are representative of the group we typically think of as 'homeless' (probably people who are rough sleeping or living in hostels) since the majority are female.

Response: We define a homeless person as an individual without permanent housing who may live on the streets; stay in a shelter, mission, single room occupancy facilities, abandoned building or vehicle; or in any other unstable or non-permanent situation. We have expanded on this area of discussion.

“It is possible that participants who reported a history of homelessness had transitioned out of homelessness many years or even decades prior. In addition, information on type of homelessness was not available. It is therefore unknown whether those who reported once being homeless were “statutory homeless”, lived on the streets, stayed in a shelter, abandoned building or vehicle, etc. Type of homelessness may have varying influences on health and behaviour. It is plausible to assume that those who are rough sleepers (living on streets, abandoned buildings or vehicles) are at a higher risk of poor health, for example, owing to exposure to cold weather and wet conditions or lack of access to essential facilities such as bathrooms. However, those who are rough sleepers are much more likely to be male (86% male) [1] and a relatively large proportion of our sample who were once homeless were female (46.7%). It may be that the present sample are not representative of the wider homeless population (or at least rough sleepers) in the UK. Future research to tease out the influence of type of previous homelessness on health/ behaviour outcomes is required.”

3. There is some discussion about the health risks of homelessness, but I think the point could be made more clearly that the difference between ex-homeless and others in this study is much smaller than between current homeless and housed people in other studies - which have found extremely

stark differences in health. This may be due to a degree of 'recovery' from homelessness, but also that the type of homelessness experienced by people in this study is different and less 'severe'.

Response: This is not the focus of the paper. Nevertheless we have expanded discussion on this.

“Interestingly, while differences in health outcomes (self-rated health, limiting long-standing illness, depressive symptoms, life satisfaction and QOL) between the present sample (ex homeless) and those who have not been homeless were significant, the magnitude of the associations was smaller than has been documented in previous studies [38]. This may be owing to a degree of ‘recovery’ from homelessness. It may also be an artefact of the type of homelessness. The majority of the present sample who had experienced homelessness may have been “statutory homeless” where health outcomes are likely better than rough sleeping.”

Reviewer: 1

Reviewer Name: Ramón Chacón Cuberos

Institution and Country: University of Granada (Spain) Please state any competing interests or state 'None declared': None declared

Please leave your comments for the authors below After reviewing the manuscript entitled "Health behaviours and mental and physical health status in older adults with a history of homelessness: a population based study", indicate that the authors have made the most suggested changes. Likewise, it is noticeable that the manuscript is well structured, written with a high quality and the pertinent statistical analyzes are used. In addition, it presents a topic of interest which is adjusted to the editorial line of the journal, so its publication is recommended.

Response: Thank you.